# Realization of Force Detection and Feedback Control for Slave Manipulator of Master/Slave Surgical Robot

**DOI:** 10.3390/s21227489

**Published:** 2021-11-11

**Authors:** Hu Shi, Boyang Zhang, Xuesong Mei, Qichun Song

**Affiliations:** 1School of Mechanical Engineering, Xi’an Jiaotong University, Xi’an 710049, China; zhangboyang@stu.xjtu.edu.cn (B.Z.); xsmei@xjtu.edu.cn (X.M.); 2Second Affiliated Hospital, Xi’an Jiaotong University, Xi’an 710004, China; qcs@xjtu.edu.cn

**Keywords:** minimally invasive surgical robot, force measurement, force feedback, impedance control, force control

## Abstract

Robot-assisted minimally invasive surgery (MIS) has received increasing attention, both in the academic field and clinical operation. Master/slave control is the most widely adopted manipulation mode for surgical robots. Thus, sensing the force of the surgical instruments located at the end of the slave manipulator through the main manipulator is critical to the operation. This study mainly addressed the force detection of the surgical instrument and force feedback control of the serial surgical robotic arm. A measurement device was developed to record the tool end force from the slave manipulator. An elastic element with an orthogonal beam structure was designed to sense the strain induced by force interactions. The relationship between the acting force and the output voltage was obtained through experiment, and the three-dimensional force output was decomposed using an extreme learning machine algorithm while considering the nonlinearity. The control of the force from the slave manipulator end was achieved. An impedance control strategy was adopted to restrict the force interaction amplitude. Modeling, simulation, and experimental verification were completed on the serial robotic manipulator platform along with virtual control in the MATLAB/Simulink software environment. The experimental results show that the measured force from the slave manipulator can provide feedback for impedance control with a delay of 0.15 s.

## 1. Introduction

Minimally invasive surgery (MIS) has the advantages of less trauma, low risk of infection, and short recovery time, and leads the development trend of clinic surgery [1]. As a result of the technology advancement in medical robotics, the master–slave type MIS robot has become an attractive research topic. When using a MIS robot in master–slave mode for surgery, doctors expect to have the same real feeling as when directly operating surgical instruments, which requires the surgical robot to be capable of force feedback at the end of surgical instruments [2]. Previous studies have shown that MIS robots with force feedback can reduce the damage to human tissue during surgery, decrease the postoperative complications, reduce the labor intensity of doctors by 30% to 60%, and lower the incidence of error during surgery by 60% [3]. Therefore, force detection and feedback are significant issues involved in MIS robot development from the aspects of both operation and rehabilitation.

The MIS robot holds surgical instruments at the end of the slave manipulator, and the surgical instruments are in contact with the patient’s tissue. Obtaining the contact force between the surgical instrument and the tissue is the premise for providing give force feedback to the doctor’s hands. Therefore, scholars in the surgical robot field have carried out investigations into force detection on surgical instruments. According to the operating principle, they can be mainly divided into resistance-based force detection, fiber-based force detection, and dislocation-based force detection. The basic idea to realize force detection in these previous research studies is to transform signals in other forms into force through calibration and decoupling. For the first detection method, electric signals were collected directly from the measured object. G S Fiscer et al. [4] developed a gripper with force detection, which is used for the clamping of abdominal tissue. The experimental results show that the gripper with three-dimensional force detection can measure the clamping force of 0–5 N, and the average measurement accuracy is less than 0.1 N. A force detection device with four degrees-of-freedom designed by U Kim et al. [5] adopts the principle of capacitive energy conversion and is used to detect axial and radial forces. The simulation results proved that the root mean square error of the measuring force of this surgical instrument is 0.1 N. The team of Tavakoli [6] developed a surgical instrument with force detection function. The clamp at the end is driven by a linear motor connected by a slender tube, and the device can measure forces or moments in six dimensions. For the detection method using fiber, the force signal is obtained indirectly. Peirs et al. [7] designed an optical force detection device for a minimally invasive surgical robot system, in which three optical fiber displacement sensors are used to measure the force. When the surgical instrument is subject to external force, the distance detected by the optical fiber sensor will change. The measuring accuracy of the device is 0.04 N. Another three-dimensional force detection device designed by Puangmali et al. [8] also adopts the principle of optical fiber detection, achieving a maximum measurement accuracy of 0.05 N. For the detection method based on deformation, the strain gauge is usually employed to sense the external force. The slender sleeve of the surgical instrument developed by Moradi et al. [9] is equipped with a strain gauge to measure the contact force between the end of the instrument and human soft tissue. It can effectively avoid the interference caused by friction between the sleeve and the rod of the surgical instrument. The force detection device designed by Zhan Juncheng [10] follows the transformation principle of strain to electricity to force. Unfortunately, this force detection device adopts linear decoupling and is easily affected by random errors, so the force detection results are unsatisfactory in the process when considering dynamics. In addition, the force detection device based on the structure of the Stewart platform was fabricated by Li Kun et al. [11], which is used to measure the force and moment in six dimensions.

Overall, the problems to be solved in force detection and feedback control of master-slave MIS robot are as follows: (1) The force detection on surgical instruments is affected by many factors, such as compactness of surgical instruments, difficulty in arrangement of force detection elements, and measurement interference. Coupling in multi-dimensional force detection affects the detection accuracy. (2) The force feedback of the surgical instrument at the end of the slave manipulator is mainly realized by measuring the output torque of the joint motor and calculating the magnitude of the force. In this way, the complexity of the control system is significantly increased. 

This study aimed to realize a simple means of force detection and feedback control for the slave manipulator of a master/slave surgical robot. The design and performance verification of the force detection device at the end of the slave manipulator are presented in Section 2. A model of the control system was built in MATLAB/Simulink based on the detected force feedback, and the contact force control was verified through virtual experiment.

## 2. Force Detection at the End of Slave Manipulator

When using the master-slave MIS robot for surgery, it is desired that the system can obtain the contact force between the surgical instrument located at the end of the slave manipulator and the lesion tissue of the patient in real time, and that the force is provided to the operator of the master manipulator, so that the operator can get an intuitive sense of the force interaction. In practice, however, it is difficult to measure the contact force directly and accurately.

### 2.1. Layout of Force Detection Elements

Strain gauges have the advantages of small size, high sensitivity, and simple measurement process, and have been widely used in the field of force detection [12]. Therefore, strain gauges were also taken as the force detection elements in this study. When detecting the force of the surgical instruments from the end of the manipulator, it is necessary to consider the arrangement of the detection elements. Generally, there are four locations for the arrangement of the elements, as shown in Figure 1.

➀ is the place where the surgical instrument is connected with the slave manipulator, and the contact force between the surgical instrument and the patient is obtained by measuring the strain at the joint with the force detection element. However, it is too far from the force acting point.

➁ is located on the axis of the surgical instrument outside the abdominal tissue, and has enough space to arrange the detection elements. However, the friction between the outer tube and the surgical instrument will greatly interfere with the detection results.

➂ is located on the axis of the surgical instruments inside the abdominal tissue, and is usually considered to be the most suitable location for detection elements. However, the layout of elements here is subject to space restrictions.

➃ is located on the side of the clamp claw, and is also a preferable location to carry out more direct force measurement. However, the claw is compact in structure, and the detection element has to be fabricated to be as small as possible.

After considering the above advantages and disadvantages, the force detecting device in this study used an arrangement of the elements along the axis of the surgical instruments inside the abdominal tissue, as in ➂ where there is less interference, the space is relatively large, and the prototype is relatively easy to produce.

### 2.2. Design of Force Detecting Device

Taking surgical procedures for benign prostatic hyperplasia by transurethral resection as an example, it is necessary to detect the force in three directions (*x*, *y*, *z*), while the torque of the other three dimensions can be ignored [13]. Therefore, a three-dimensional force detection device was designed in this study. The force detection element needs to be attached to the elastomer, so the elastic deformation of the elastomer determines the sensitivity of the force-sensing device. The elastomer element in this study was designed as an orthogonal beam structure, which is composed of three groups of orthogonal beam structures as part of the axis of the surgical instrument. The schematic diagram is shown in Figure 2. The advantage of using the orthogonal beam structure is that when there is contact force between the surgical instrument and the patient, the three groups of orthogonal beams will be deformed at the same time. Strain gauges are affixed to three groups of orthogonal beam structures, so the force signals can be collected by measuring the resistance changes of the strain gauges.

Compared with the tubular structure, the elastomer element with an orthogonal beam structure has higher sensitivity in all directions. As shown in Figure 3, each group of orthogonal beams is composed of three parts: the middle part is made of a steel sheet, and the other two parts are fixed to the steel sheet by rivets. The steel sheet is made of 65 Mn spring steel, and the rest is made of nylon with good strength and elasticity.

Taking into account the force detection during operation, the maximum strain of steel sheet is 1 mm, the width of steel sheet is 5 mm, the length of crossbeam is 3 mm, and the length of the cantilever beam is 18 mm. The final force detection device prototype is shown in Figure 4a. The outer diameter of the force detection device is 14 mm, the overall length is 90 mm, the maximum axial loading force is 8 N, and the maximum radial loading force is 6 N.

To further discuss the applicability of the force detection device proposed above, surgery for benign prostatic hyperplasia (BPH) treatment was taken as an example to analyze the performance of surgical instrument. As shown in Figure 4b, the force detection device that was designed to fit the surgical instrument looks like a tube. When it is put into operation, the device can be embedded to form a section of the surgical tool. 

### 2.3. Calibration of Force Detecting Device

The device has three sets of output voltage during force measurement, which need to be converted into force in three directions. First, the static calibration needs to be carried out by experiment, and the standard weight is used to simulate the load. The specific calibration process is described as follows:(1)Increase the weight in the positive direction of the x-axis gradually up to 600 g, and record the corresponding voltage value.(2)Gradually remove the weight in the positive direction of the x-axis, and write down the voltage value after each step.(3)Repeat the loading and unloading process four times, take the average value, and obtain the relationship between the loading force and the output voltage in the positive direction of *x*.(4)Change the force direction so that the load is negative along the *x*-axis, and repeat steps (1)–(3).(5)The calibration process of the *y*-axis direction is performed in the same way as for the *x*-axis direction. For the *z*-axis direction, only compression along the axis is exerted, and other steps are the same as for the x-axis calibration process.

It should be noted that before starting the calibration experiment, it is necessary to set values to zero so that the output voltage of the differential amplifier circuit is 0 V. Because several groups of differential amplifying circuits work at the same time, the microcomputer can obtain readings successfully regardless of whether the output voltage of the Wheatstone bridge is positive or negative.

The results obtained in the calibration experiment are shown in Figure 5. Figure 5a–c shows the calibration results in three directions, respectively, and Figure 5d depicts the direction of the loading force. From Figure 5a–c, it can be seen that the output voltage proportionally changes with the force loading, and a good linear relationship between input and output was achieved. According to the results obtained from four measurements, the repeatability for the error of the force detection device in three directions is 5.2%.

### 2.4. Output Force Decoupling

After calibration, it is necessary to determine the corresponding relationship between the output force and the output voltage in three directions. Considering the influence of nonlinear factors, the extreme learning machine (ELM) algorithm is selected to decouple the signal into three directions. Basically, ELM operates like a neural network, of which the structure is composed of an input layer, output layer, and hidden layer, as shown in Figure 6. When decoupling the above calibration results, there are three neurons in each of the input and output layer. Taking into account the time assumption and accuracy, the neuron number *k* was chosen as 25.

The decoupling calculation was carried out by coding in MATLAB software. The voltage data in the calibration data set is [*u*_1_ *u*_2_ *u*_3_], and the force data is *F_x_*_,*y*,*z*_. The voltage data are given to the ELM, and the final force output by prediction is *f_x_*_,*y*,*z*_. The predicted result was compared with the given data to verify the decoupling result. The loading forces in the *x*, *y*, and *z* directions obtained through decoupling are shown in Figure 7.

Figure 7a–c shows the decoupling results after applying forces in the *x*, *y*, and *z* directions, respectively. Figure 7d–f shows the error rate of the decoupling results corresponding to the different components. The error rate is expressed as follows:(1)$$\mathrm{error}\;\mathrm{rate}=\left|\frac{F_{x,y,z}-f_{x,y,z}}{F_{f}}\right|$$
where *F_f_* is the full range value, and *f_x_*_,*y*,*z*_ is the output force after decoupling. When the force is applied in a certain direction, the error rate in the corresponding direction is called the measurement error rate, which reflects the measurement accuracy, whereas the error rate in other directions can be called the coupling error rate, and reflects the coupling relations between this direction and other directions. Figure 7d–f show that the maximum measurement error rates of the force measuring devices in the *x*, *y*, and *z* directions are 2.21%, 2.52%, and 6.27%, respectively. The coupling of *F_x_* to *F_y_* and *F_z_* is small, and the maximum coupling error rate is 1.96% and 3.56%, respectively. The maximum coupling error rate of *F_y_* to *F_x_* and *F_z_* is 1.96% and 3.56%, respectively. The coupling of *F_z_* to *F_x_* and *F_y_* is very small, and the maximum coupling error rate is 1.42% and 0.98%, respectively. The results indicate the measurement accuracy of the force detecting device can meet the requirements of a surgical instrument. In addition, it should be noted that the error rate of *F_z_* decoupling in other directions is not as stable as that of the other components. The *z* direction represents the axis of the tube, and a strain gauge installed in this manner is not sensitive to the sensing force. In practice, the force action along the axis is not frequently required. 

### 2.5. Force Detecting Performance Verification

In surgery, the tool–tissue interaction force acting on the surgical instrument changes dynamically, so tests under varying conditions must be carried out to verify the performance of the force detection device. In this study, the HZC-T high-precision column sensor was adopted to provide a comparison with the designed force detection device for verification. Its operating range is 0–10 N, and the measurement accuracy is 0.05%. The test process is shown in Figure 8. The force detection device is fixed and the dynamic performance was verified by loading random forces to the reference force sensor. The experimental results are shown in Figure 9. As shown in the figure, the variation trend in the force detected by the force detection device is consistent with that of the reference force sensor, but there exists a certain hysteresis. The main reason for this is that when the elastic element of the force detection device deforms, it takes a certain time to transfer the deformation to the strain gauge on the 65 Mn steel sheet.

Analyzing the above results obtained during the whole detection process, the average detection error of the force detection device in the *x*, *y*, and *z* directions is 0.38, 0.31, and 0.42 N, and the root mean square error is 0.62, 0.45, and 0.59 N, respectively. The error rates in the three directions are 9.5%, 7.7%, and 10.5%, respectively. Because the force feedback of the surgical robot can be scaled to improve hand feeling, satisfactory accuracy of the force detection device is achieved in the process of dynamic measurement. In addition, the resolution of the force sensing device is 0.01 N, which is high enough to detect the interactions in the surgical scenario.

## 3. Force Control at the End of Slave Manipulator

Because the force sensing results from the end of slave manipulator are finally used to regulate the force interactions between the surgical instrument and human tissue, force feedback and control should be taken into consideration. Safety is the priority in surgery, so the interaction force between the robotic manipulator and tissue must be restricted within a certain range. In order to investigate the force control performance, a serial robotic arm with five rotary joints, as shown in Figure 10, was built to operate as the slave manipulator together with the microcomputer as the master [14].

### 3.1. The Control Strategy

During the operation, the contact force between the surgical instruments at the end of the slave manipulator and the human tissue must be strictly controlled to ensure that the patient will not be harmed by incorrect operation. For the force control of the slave manipulator in tele-operation, this issue can be addressed by implementing adaptive control, force/position hybrid control, impedance control, etc. [15,16]. The complexity of the control system is critical to the robotic operation characteristics. For the force control with respect to surface contact, the impedance control strategy is widely adopted by industrial or healthcare robots. It has been proved to be suitable for handling the human interaction issues. Because the surgical tool usually moves slowly during surgery and the human tissue is so soft, the interacting force may not experience a dramatic change [17]. In this research, a position-based impedance control method is proposed to control the force from the end of the manipulator. For impedance control, the manipulator and contact force are regarded as a spring-damping-mass model. Therefore, the force/position relationship of the manipulator can be expressed as:(2)$$M_{d}\left({\ddot{x}}_{c}-{\ddot{x}}_{d}\right)+B_{d}\left({\dot{x}}_{c}-{\dot{x}}_{d}\right)+K_{d}\left(x_{c}-x_{d}\right)=F_{e}$$
where *x_c_* is the actual position of the end of the slave manipulator; *x_d_* is the desired position of the end of the slave manipulator; *M_d_* is the mass coefficient; *B_d_* is the damping coefficient; *K_d_* is the stiffness coefficient. Defining the correction *e* = *x_d_* − *x_c_*, Equation (2) can be expressed as:(3)$$M_{d}\left(\ddot{e}\right)+B_{d}\left(\dot{e}\right)+K_{d}\left(e\right)=-F_{e}$$

Writing into the transfer function in the frequency domain, Equation (3) can be expressed as:(4)$$e(s)=\frac{-F_{e}(s)}{M_{d}s^{2}+B_{d}s+K_{d}}$$

Once the position correction *e* is obtained, the actual position *x_c_* can be obtained by adding the desired position *x_d_*, and then the angular position *θ_c_* of each joint motor is calculated according to the inverse kinematics equations. Finally, this is delivered to the controller to control the position of the manipulator, so that the contact force is always kept within the allowable range.

### 3.2. The Process of Force Control

This research aimed to achieve the force control for a master–slave MIS robot, so the action of the slave manipulator is controlled by the master manipulator. The force control process involved in this research is shown in Figure 11. When the force is collected from the force detection device at the end of the manipulator, the master manipulator will feedback the force in real time so that the operator can feel the presence of force. If the detected force is less than the set value, the slave manipulator will follow the master manipulator synchronously, and force control is inactive. If the detected force is greater than the set value due to unexpected reasons, the slave manipulator will carry out impedance control, and a light or beep is triggered to inform the operator.

### 3.3. Control System Modeling

In this study, the force control system modeling was conducted in the MATLAB/Simulink software environment, which can provide a visual control platform for the subsequent simulation and experiment. As shown in Figure 12, this mainly includes a force signal module, a master manipulator module, a kinematics module, a reset module, a dynamics module, an impedance control module, and a signal output module.

Figure 13 illustrates the principle of the impedance control module in detail. In the process of impedance control, the magnitude of the position correction *e* is determined by the actual detecting force *F_c_* and the expected maximum contact force *F_d_*. The maximum position correction is set to 0.1 rad, and the slave manipulator can smoothly complete the impedance control process by continuously changing the parameter. Therefore, the force control is realized by changing the position of the end of the slave manipulator correspondingly. On pressing the reset button, the impedance control module will calculate the position correction *e* according to the current position of the slave manipulator and the master manipulator, and make the slave manipulator move smoothly and quickly by changing *e*. Finally, it maintains the same attitude as the master manipulator; that is, the master–slave cooperative motion remains and the impedance control process is achieved.

The block diagram of the control system modeling in MATLAB/Simulink is shown in Figure 14. The force signal module outputs the detecting force to the impedance control module. If the detecting force is less than the set value, the impedance control module will directly transfer the position signal collected by the master manipulator signal module to the kinematics module. The joint angles solved by the kinematics module are transmitted to the dynamic module and the signal output module, so that the slave manipulator and the master manipulator work together. If the detecting force is greater than the set value, impedance control will take effect. First, the impedance control module will add the position correction *e* to the initial expected position to obtain the actual position, and then transfer the actual position to the kinematics module. After solving, the position control quantity of each joint motor is obtained. Finally, the position control command is given to the dynamic module and the signal output module, so that the slave manipulator can carry out position-based impedance control. Under this circumstance, the surgical instruments at the end of the slave manipulator will disconnect from the lesion tissue and return to an idle position. Therefore, the force detected at this position is less than the set value, which can achieve the contact force control at the end of the slave manipulator.

Under the circumstance of impedance control, the operator of the master manipulator will notice the indicator light, and change the position and status of the surgical instruments at the end of the slave manipulator. Because the master–slave cooperative motion is suspended at this moment, the operator must press the reset button in order to start the master–slave cooperative motion. At the same time, the impedance control module will reconfirm whether the detecting force is less than the set value. If this is true, the master–slave control mode will be restored, otherwise the impedance control will continue. For the system recovery, the slave manipulator will undergo a fast following process. When the master and slave manipulator reach the same position and posture by self-adjustment, the master–slave cooperative motion is ready for operation.

### 3.4. Simulation and Experiment

Based on the models created in MATLAB/Simulink software, the simulation and the experiment can be carried out simultaneously. The virtual model takes control of the motion of the slave manipulator according to the position signal of the master manipulator. During operation, the force detection device at the end of the slave manipulator transmits the force signal into the model in the computer in real time. The master end generates forces through a rotational potentiometer regulated by a handle. If the force is greater than the set value, it will stop the master–slave cooperative motion and the impedance controller takes effect. Throughout the process, the MATLAB software reads the position signals of the joint motors of the slave manipulator. After pressing the reset button, if the force signal is less than the set value, the model will quickly restore the master–slave cooperative control by following instructions.

The duration of the model verification process is 10 s, the sampling frequency is 100 Hz, and the reset signal is given at 6.5 s. The position and attitude signals of the master manipulator are input in the form of a data table. For the convenience of analysis and observation, the yaw angle *γ* of the surgical instrument is kept constant as π/12, and the pitch angle *β* tracks a sinusoidal curve, as shown in Figure 15. The expected trajectory of the end of the slave manipulator rotates clockwise for the first 5 s, and the maximum rotation angle is 5.236 rad. After 5 s, it rotates counterclockwise and returns to the original position along the given trajectory.

During the model verification, the force signal is set at 2.5 N. The axial force is applied to the force detection device at the end of the slave manipulator at a certain moment between 3 and 5 s, so that the axial force is greater than the set value. The force signal module receives the force signal to decouple, and the axial force *F_z_* after decoupling is shown in Figure 16. It can be seen that the detected force is greater than the set value at 4.4 s. After that, a reset signal is given at 6.5 s to verify the process of recalling the master–slave control action.

In order to study the performance of position-based impedance control, the joint angle of robotic arm calculated by the kinematics module should be considered. The desired output angle was compared with the angle calculated by simulation model, and the impedance control effect was analyzed. The angle output of each joint is shown in Figure 17. Here, only joints 1, 2, 4, and 5 are considered because the angle of joint 3 of the slave manipulator changes very little [14]. It can be seen that the expected output angle is consistent with the output angle in simulation during the period of 0–4.51 s. At 4.52 s, the master–slave control action is cut off, and the slave manipulator begins to carry out impedance control. As a result, the output angle in the simulation is not consistent with the desired output angle between 4.52 and 7.80 s. In the process of impedance control, the output angle of each joint in the simulation between 4.52 and 6.31 s is dependent on the position at 4.52 s, and the position of the slave manipulator is obtained by changing the position correction *e*. At 6.31 s, the surgical instrument at the end of the slave manipulator disconnects from the lesion tissue and returns to a safe position. Between 6.31 and 6.5 s, the impedance control process is completed, and the joint angle of the slave manipulator remains unchanged, so the position and posture of the end of the slave manipulator holds. At 6.5 s, the reset is activated to restore the master–slave control mode. Between 6.5 and 7.80 s, the slave manipulator is subject to constantly changing the amount of position correction to achieve the fast following process, and finally the position of the end of the slave manipulator is consistent with that of the master manipulator. In order to prevent shaking of the slave manipulator, the trajectory during this period is planned with a smooth polynomial so that the angular velocity of each joint motor gradually increases from zero. Starting from the moment at 7.81 s, the master–slave cooperative movement is completely recovered.

The position of each joint motor of the slave manipulator measured in real time was compared with the simulation results of the kinematics module, as shown in Figure 18. It can be seen that the angles measured by the experiments of joints 4 and 5 are basically the same as those of the simulation output, and the average errors are 0.0021 and 0.0018 rad, respectively. The average errors of joints 1 and 2 were 0.0042 and 0.0103 rad, respectively. Because the communication between the computer and the slave manipulator leads to a time delay of 0.04 s, when the contact force is greater than the set value, the slave manipulator will begin to carry out impedance control after 0.05 s. Joint 2 is located at the bottom, and nearly carries the weight of the whole manipulator, so the output angle of joint 2 in experiment is slightly larger than that in the simulation. Joint 1 is rigidly attached to the platform base, so its error mainly results from the machining accuracy of parts and assembly. It should be noted that between 6.5 and 7.8 s, the slave manipulator is in a fast following stage, and the angle signal read by each joint motor is very stable in this process. This proves that there is no obvious shaking in the slave manipulator during the period of restoring the master–slave control mode.

The force control process also can be reflected by the trajectory of the end of the slave manipulator, as shown in Figure 19. The expected trajectory is plotted according to the data from the master manipulator, whereas the simulation trajectory was calculated in the model. The difference between the two lines is dramatic at the stage of force control implementation. When the detecting force at the end of the slave manipulator is less than the set value, the trajectory at the end of the slave manipulator is consistent with the desired trajectory. When the contact force is greater than the set value, the control system cuts off the master–slave control, and the slave manipulator carries out impedance control. As a result, the end force control is realized by modifying the position control quantity of the slave manipulator. After pressing the reset button, the slave manipulator restores master–slave control and the trajectory follows quickly. The experimental results show that the actual movement of each joint of the manipulator is consistent with that obtained through the simulation model, and the joint motion has better following performance regardless of the contact force variation.

In robot-assisted surgery, the doctor holds the handle at the master level of the robotic system. Thus, the process is a typical type of tele-operation. The main objective of the control scheme proposed above is to effectively avoid dangerous movement, which may exert an overload onto human tissue, with simple implementation of a control action. Compared with a control system with a complicated algorithm, the proposed approach can more easily achieve real-time regulation, although the accuracy is relatively lower. However, in this study we mainly focused on danger alarms and protection, and the force control accuracy in surgery is more dependent on the operator, and is not as demanding as in the case of an industrial robotic arm.

## 4. Conclusions

This study addressed the force detection and feedback control implementation at the end of the slave manipulator for a master/slave MIS robot. The force detection device was developed by integrating three orthogonally deployed strain gauges into a tubular surgical instrument. The force sensing ability in three directions was tested by decoupling with the ELM method. The calibration and decoupling experiment showed that the force detection device can achieve high measuring accuracy for the load applied in different directions. Based on the detected force of the slave manipulator, the force control was studied to provide a safe operation condition, considering the interactions between surgical instrument and human body tissue. The control objective was modeled using a computer and a simulation connected with hardware was conducted to verify the impedance control strategy. It was proven that the physical slave manipulator can perform the motion by closely following the instructions from the simulation model, which acts as the master manipulator, and force feedback control was realized under different conditions. The research results provide a possible solution for the force sensing and handling of master/slave MIS robots applied in surgery.

## Figures and Tables

**Figure 1 sensors-21-07489-f001:**
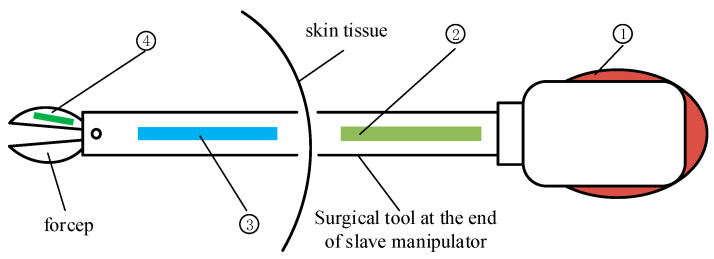
Layout scheme of force-sensing elements.

**Figure 2 sensors-21-07489-f002:**
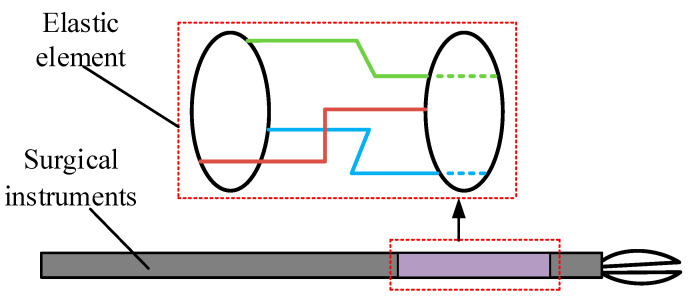
Elastomer element structure diagram.

**Figure 3 sensors-21-07489-f003:**
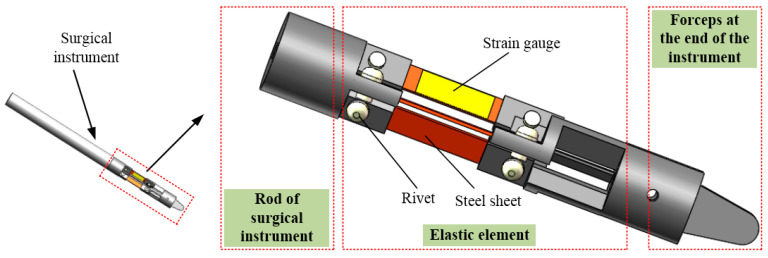
Three-dimensional model of elastic structure.

**Figure 4 sensors-21-07489-f004:**
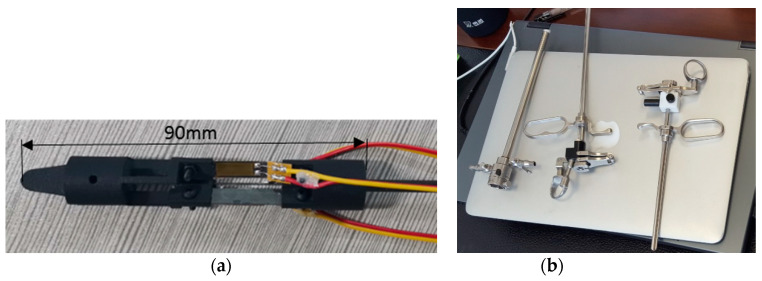
Prototype of the force detection device. (**a**) force detection device to be integrated, (**b**) surgical instrument in application.

**Figure 5 sensors-21-07489-f005:**
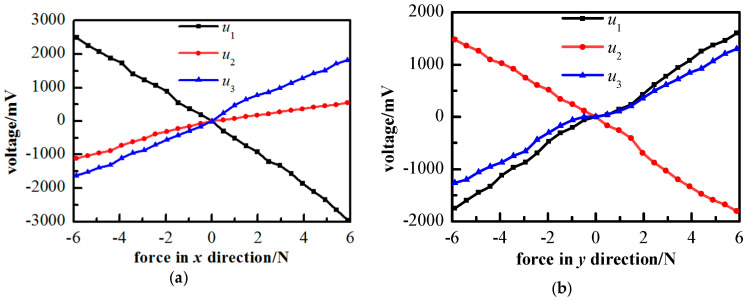
Calibration result of the force detection device: (**a**) calibration result of *x*-axis, (**b**) calibration result of *y*-axis, (**c**) calibration result of *z*-axis, (**d**) schematic diagram of the direction of the loading force.

**Figure 6 sensors-21-07489-f006:**
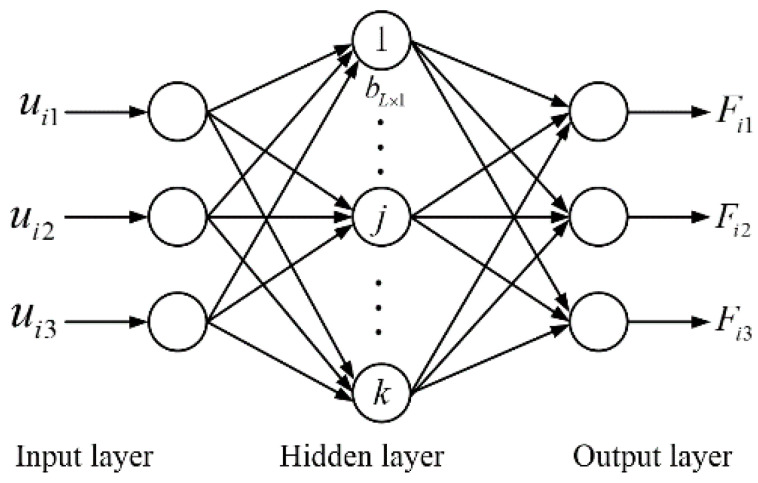
Network architecture based on ELM.

**Figure 7 sensors-21-07489-f007:**
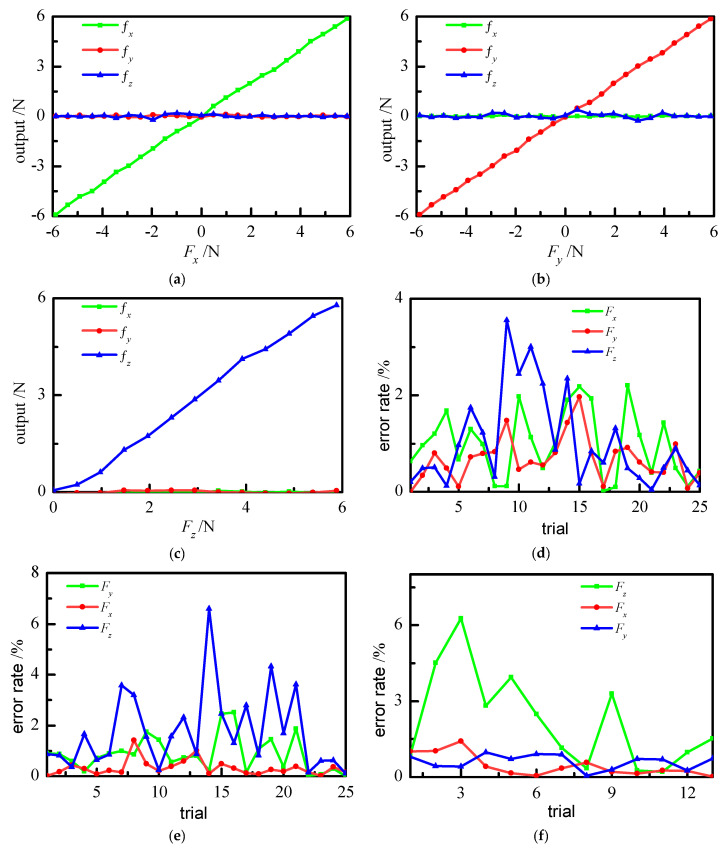
Decoupling result of force output based on ELM: (**a**) decoupling results of *F_x_*, (**b**) decoupling results of *F_y_*, (**c**) decoupling results of *F_z_*, (**d**) error rate in the x-direction decoupling, (**e**) error rate in the y-direction decoupling, (**f**) error rate in the z-direction decoupling.

**Figure 8 sensors-21-07489-f008:**
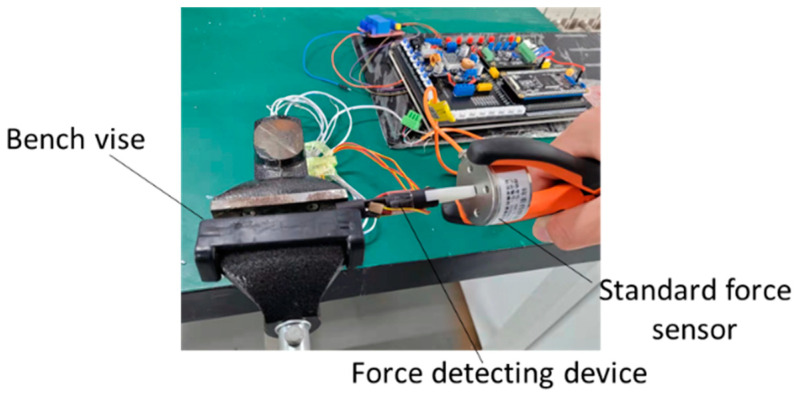
Scene of experimental verification for force detecting performance.

**Figure 9 sensors-21-07489-f009:**
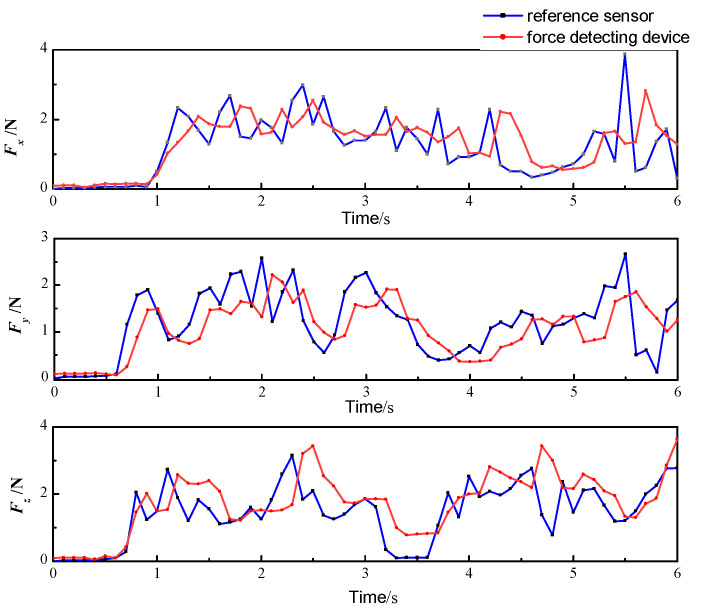
Dynamic performance verification results of the force detection device.

**Figure 10 sensors-21-07489-f010:**
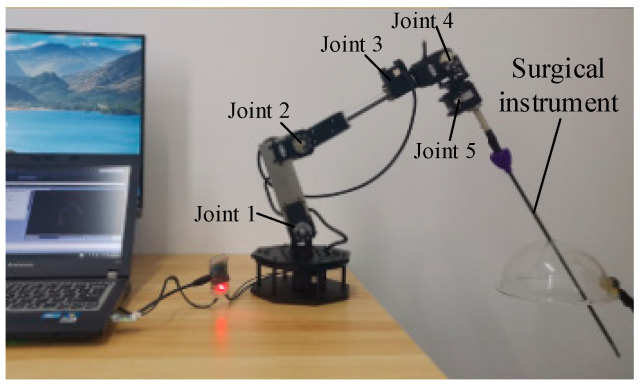
Slave manipulator used for force control implementation.

**Figure 11 sensors-21-07489-f011:**
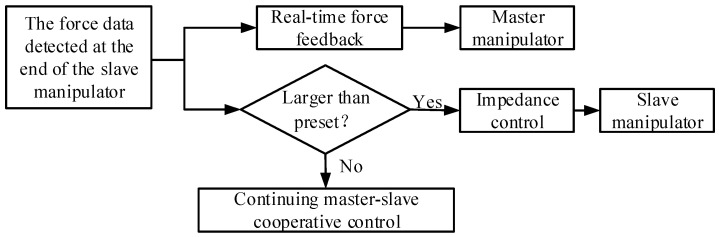
Flowchart of force control procedure.

**Figure 12 sensors-21-07489-f012:**
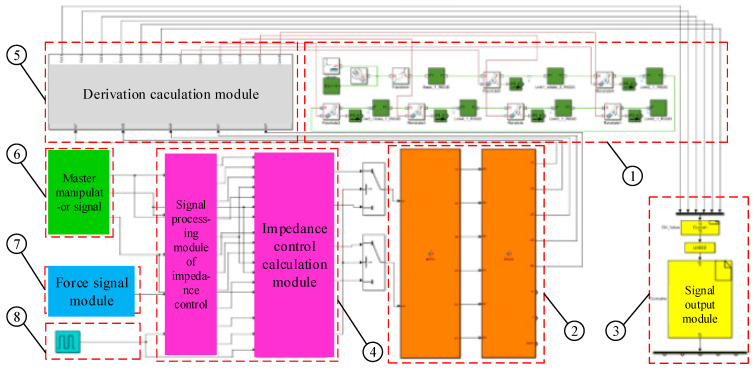
Control system models built in MATLAB/Simulink: 1. dynamics module; 2. kinematics module; 3. signal output module; 4. impedance control module; 5. derivation calculation module; 6. master manipulator signal module; 7. force signal module; 8. reset module.

**Figure 13 sensors-21-07489-f013:**
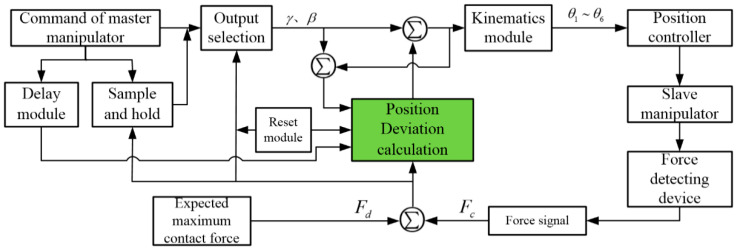
Working principle of the impedance control module.

**Figure 14 sensors-21-07489-f014:**
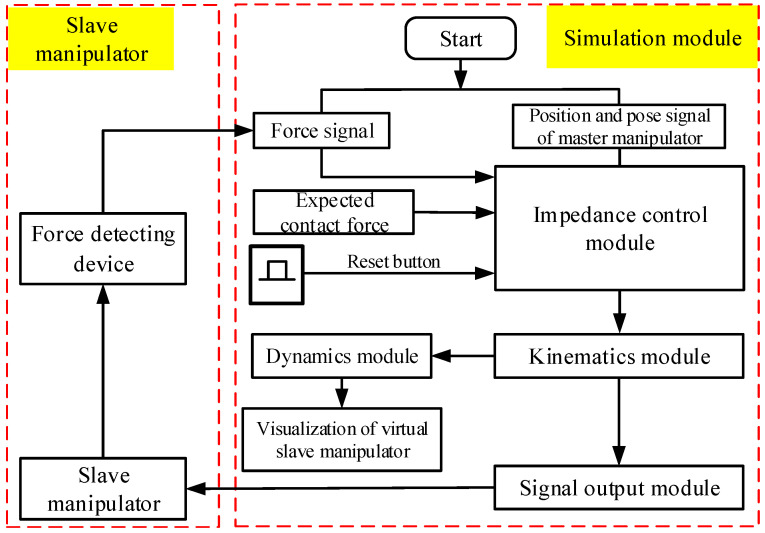
Block diagram of the working principle of the control system model.

**Figure 15 sensors-21-07489-f015:**
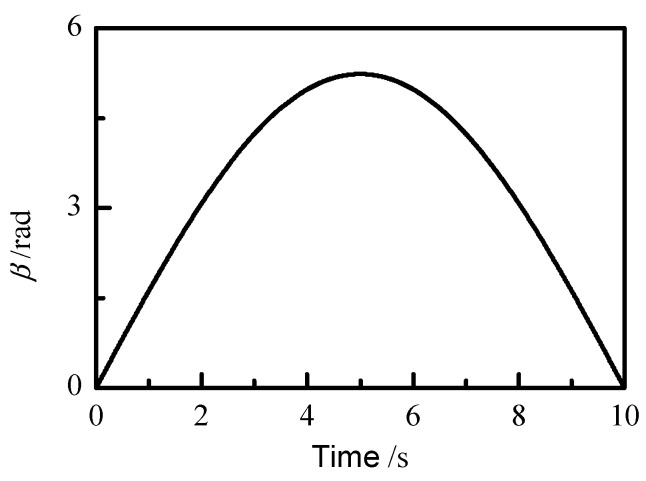
Pitch angle input.

**Figure 16 sensors-21-07489-f016:**
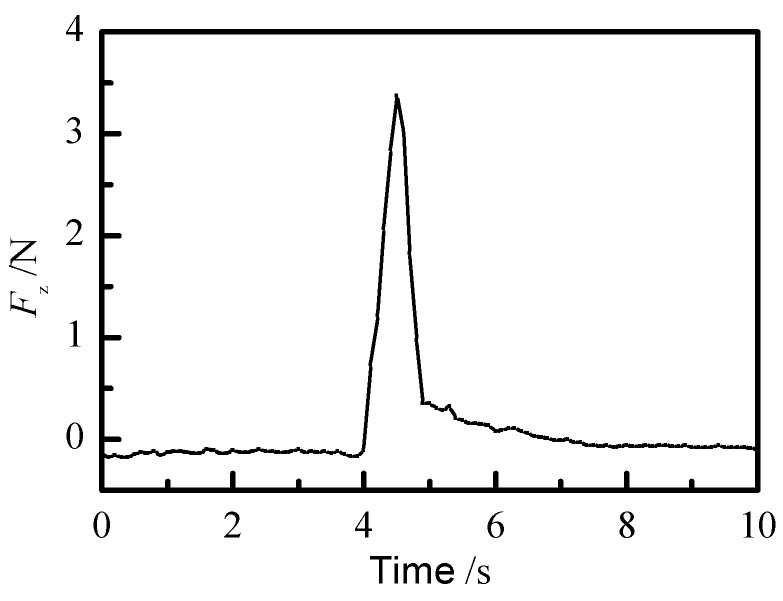
Force signal tracks in simulation.

**Figure 17 sensors-21-07489-f017:**
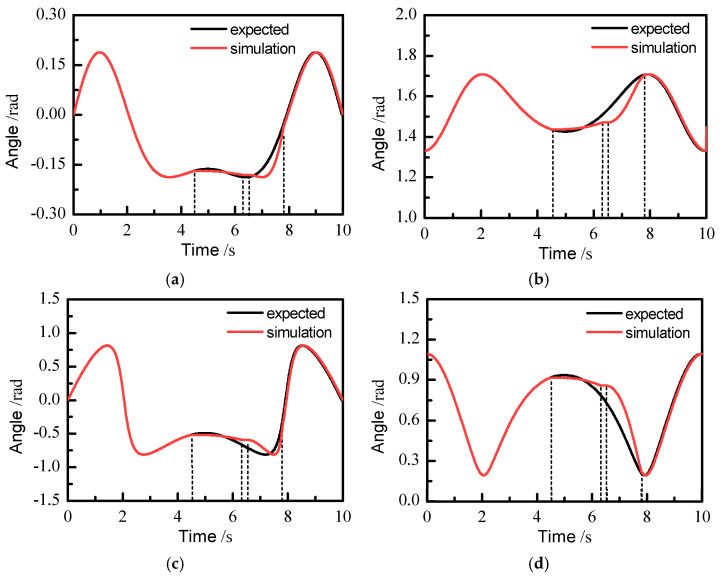
Output joint angles in the simulation model: (**a**) angle of joint 1, (**b**) angle of joint 2, (**c**) angle of joint 4, (**d**) angle of joint 5.

**Figure 18 sensors-21-07489-f018:**
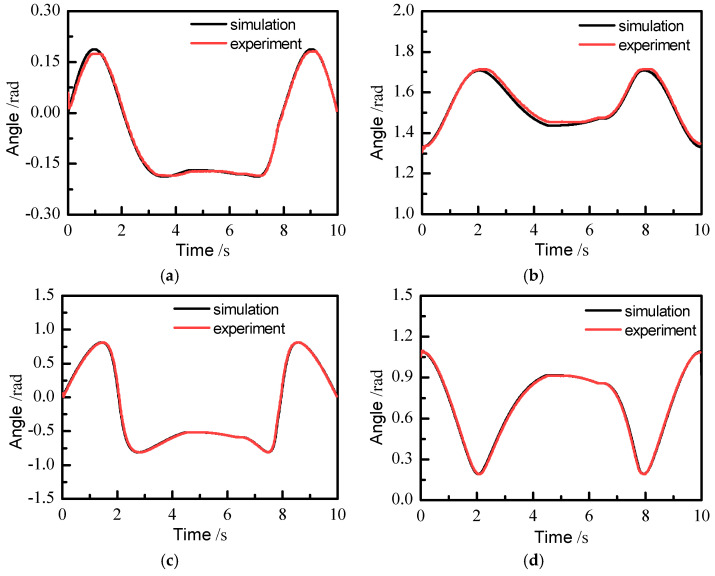
Comparison between the experimental and simulation results of each joint motor: (**a**) output angle of joint 1, (**b**) output angle of joint 2, (**c**) output angle of joint 4, (**d**) output angle of joint 5.

**Figure 19 sensors-21-07489-f019:**
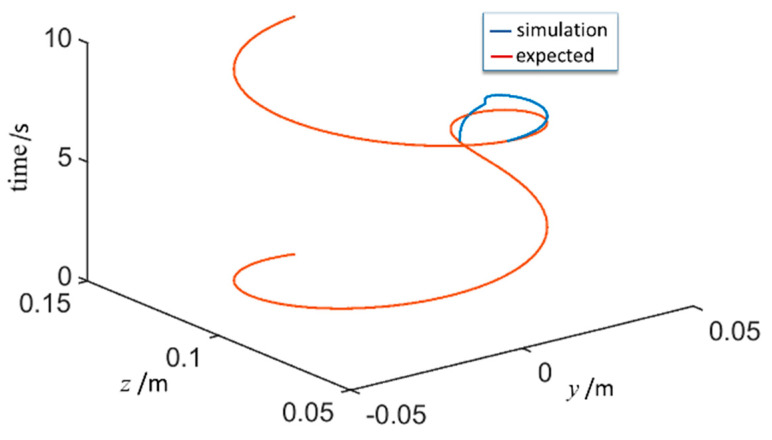
Trajectory of the end of the slave manipulator.

## Data Availability

Not applicable.

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
