# Peer review of "Realization of Force Detection and Feedback Control for Slave Manipulator of Master/Slave Surgical Robot"

_sensors, 2021, doi:10.3390/s21227489_

Round 1

Reviewer 1 Report

This paper mainly deals with the force sensing and feedback of surgical instrument. The force detecting devices was developed and integrated into the surgical instrument for compactness. Based on the force results obtained by the measuring device, impedance control of interactions between surgical instrument and human tissue was verified through hardware in loop simulation. The topic is very important to the surgical robot and the research results are valuable. The manuscript is written in an easily understood manner. Comments are given as follows.
1.       The direction of Fz in Fig.5 can be adjusted oppositely, because stretch is usually taken as positive force in aspects of mechanics.
2.       Can the force decoupling section be described in detail about how to build the model and solve the equations?

Author Response

Dear  reviewers,

We deeply appreciate the time and effort you’ve spent in reviewing our manuscript (ID: sensors-1378673) entitled "Realization of force detection and feedback control for slave manipulator of master/slave surgical robot". Those comments are all valuable and very helpful for revising and improving our article, as well as the important guiding significance to our research next step. We have read the comments carefully and made corrections which we hope to meet with approval. The main corrections in the paper and the response to the reviewer’s comments are presented as follows.

Comments: This paper mainly deals with the force sensing and feedback of surgical in-strument. The force detecting devices was developed and integrated into the surgical instrument for compactness. Based on the force results obtained by the measuring device, impedance control of interactions between surgical instrument and human tissue was verified through hardware in loop simulation. The topic is very important to the surgical robot and the research results are valuable. The manuscript is written in an easily under-stood manner. Comments are given as follows.

  1. The direction of Fz in Fig.5 can be adjusted oppositely, because stretch is usually taken as positive force in aspects of mechanics.

Response: Fig.5(d) was revised.

  1. Can the force decoupling section be described in detail about how to build the model and solve the equations?

Response: Descriptions are added in Section 4.2.

Reviewer 2 Report

Hello, this manuscript presented a simple solution for force measurements in surgical manipulator that lacks the novelty in design and limited characterizations. The authors need to minituarize the sensor area (not in the diameter of 14 mm with limited space in the center) and report the hysteresis, accuracy, resolution, repeatability etc also. It's recommended to simplify the cotroller diagrams as they're not novel to this field. 

Author Response

Dear reviewers,

We deeply appreciate the time and effort you’ve spent in reviewing our manuscript (ID: sensors-1378673) entitled "Realization of force detection and feedback control for slave manipulator of master/slave surgical robot". Those comments are all valuable and very helpful for revising and improving our article, as well as the important guiding significance to our research next step. We have read the comments carefully and made corrections which we hope to meet with approval. The main corrections in the paper and the response to the reviewer’s comments are presented as follows.

Response: The main contribution of this paper is presenting a method to easily realize the force measurement in surgical manipulator. The cross section area is determined according to the real surgical instrument for Benign Prostatic Hyperplasia. The characteristics of the sensor are illustrated in Fig.7 and Fig.9. The resolution is added in the manuscript. As for hysteresis, it is difficult to express through a certain value since the force signal is applied randomly. The control system diagram was removed to achieve simplification.

Reviewer 3 Report

The manuscript presents a design of a slave manipulator for force detection and a system flow of feedback controls for master/slave surgical robots. The manuscript is well written and technically sound. There are a few comments for authors to consider for improvement presented below:

  1. lines 45-71: it is not easy to associate the mentioned literature with the aforementioned three means for force detections. Can the authors improve this literature review section?
  2. lines 72-84: this paragraph may be separated into two paragraphs, with the second paragraph devoted to a paper structure.
  3. "real-timely" may be changed to "in real-time". Consider this change for all occurrence.
  4. line 88: the force is (provided) to the operator...
  5. lines 113-116: please also mention the section number, 1,2,3 or4.
  6. Fig. 3: what type of surgical instruments/tools can the designed structure equip? and also for what MIS procedures? Please discusse on this.
  7. Fig. 4: can the authors display dimensions in the figure?
  8. line 175: it is not clear to the reviewer how the output force is rendered at the master side. See doi: 10.1109/RBME.2017.2773521, can the authors elaborate on the haptic rendering which requires forces to be updated at least 500-1000Hz?
  9. Following above, can the authors also discuss the tool-tissue interactions, soft tissue deformations/modelling, tissue characterisation, haptic rendering, and safety of medical robotics in robotic MIS with related references? Perhaps in the discussion section.
  10. Fig. 6: can the authors elaborate on the fluctuation of the error rates?
  11. Fig. 7: can the authors provide a picture of the actual experiment?
  12. lines 217-221: can the authors also show the errors in percentage?
  13. A comprehensive discussion section is needed, discussing various aspects of force feedback and control in MIS, and the advantages and limitations of the proposed design.

Author Response

Dear  reviewers,

We deeply appreciate the time and effort you’ve spent in reviewing our manuscript (ID: sensors-1378673) entitled "Realization of force detection and feedback control for slave manipulator of master/slave surgical robot". Those comments are all valuable and very helpful for revising and improving our article, as well as the important guiding significance to our research next step. We have made corrections which we hope to meet with approval. The main corrections in the paper and the response to the reviewer’s comments are presented as follows.

Comments:

The manuscript presents a design of a slave manipulator for force detection and a system flow of feedback controls for master/slave surgical robots. The manuscript is well written and technically sound. There are a few comments for authors to consider for improvement presented below:

  1. lines 45-71: it is not easy to associate the mentioned literature with the aforementioned three means for force detections. Can the authors improve this literature review section?

Response: The paragraph was improved.

  1. lines 72-84: this paragraph may be separated into two paragraphs, with the second paragraph devoted to a paper structure.

Response: The paragraph was separated.

  1. "real-timely" may be changed to "in real-time". Consider this change for all occurrence.

Response: It was changed throughout the text.

  1. line 88: the force is (provided) to the operator...

Response: It was revised.

  1. lines 113-116: please also mention the section number, 1,2,3 or4.

Response: It was revised.

  1. Fig. 3: what type of surgical instruments/tools can the designed structure equip? and also for what MIS procedures? Please discusse on this.

Response: It was revised.

  1. Fig. 4: can the authors display dimensions in the figure?

Response: It was revised.

  1. line 175: it is not clear to the reviewer how the output force is rendered at the master side. See doi: 10.1109/RBME.2017.2773521, can the authors elaborate on the haptic rendering which requires forces to be updated at least 500-1000Hz?

Response: Sorry, we can not find the issue described in line 175 or any other position of the manuscript.

  1. Following above, can the authors also discuss the tool-tissue interactions, soft tissue deformations/modelling, tissue characterisation, haptic rendering, and safety of medical robotics in robotic MIS with related references? Perhaps in the discussion section.

Response: It was revised.

  1. Fig. 6: can the authors elaborate on the fluctuation of the error rates?

Response: It was revised.

  1. Fig. 7: can the authors provide a picture of the actual experiment?

Response: The picture of the actual experiment can not express the principle clearly, so we choose this diagram to explain.

  1. lines 217-221: can the authors also show the errors in percentage?

Response: It was revised.

  1. A comprehensive discussion section is needed, discussing various aspects of force feedback and control in MIS, and the advantages and limitations of the proposed design.

Response: The contents are described.

Round 2

Reviewer 2 Report

the responses are not adequete

Author Response

section 2 was revised in response to the previous comments.

Reviewer 3 Report

The authors have greatly improved the manuscript, with a few minor comments remaining:

Fig. 4: there seems to be a typo at line 152, fig.14(b)->fig.4(b) and In fact->in fact.

Previous comments:

  1. "line 175: it is not clear to the reviewer how the output force is rendered at the master side. See doi: 10.1109/RBME.2017.2773521, can the authors elaborate on the haptic rendering which requires forces to be updated at least 500-1000Hz?" Sorry I did not make this clear. This comment was made at line 175, but did not apply only to this line. Instead, it raises a general question to the authors of how forces at the master side are rendered. Please elaborate it in Section 2.4 or anywhere in the manuscript where appropriate, by referring to the provided reference. 
  2. Fig. 8, can the authors provide a picture of the actual experiment alongside the illustration?

I recommend the manuscript for publication, pending the above modifications.

Author Response

  1. typo at line 152 was corrected.
  2. the explaination was added in section 3.4, and the reference was cited.

     3. Fig. 8 was replaced.